# Health and Economic Impact of Atrial Fibrillation of Workers in Italy: Social Security Benefits

**DOI:** 10.3390/ijerph19031883

**Published:** 2022-02-08

**Authors:** Marco Trabucco Aurilio, Francesco Saverio Mennini, Claudia Nardone, Andrea Piccioni, Matteo Bolcato, Vincenzo Russo, Valerio Sciannamea, Raffaele Migliorini, Luca Coppeta, Andrea Magrini

**Affiliations:** 1Department of Medicine and Health Sciences “V. Tiberio”, University of Molise, 86100 Campobasso, Italy; marco.trabuccoaurilio@unimol.it; 2CEIS EEHTA, DEF Department, Faculty of Economics, University of Rome “Tor Vergata”, 00133 Rome, Italy; mennini@uniroma2.it (F.S.M.); claudia.nardone@uniroma2.it (C.N.); 3Emergency Medicine Department, Fondazione Policlinico Universitario “A. Gemelli”, IRCCS, 00168 Rome, Italy; andrea.piccioni@policlinicogemelli.it; 4Legal Medicine, University of Padua, 35121 Padua, Italy; 5Department of Translational Medical Sciences, University of Campania “Luigi Vanvitelli” Monaldi Hospital, 80131 Naples, Italy; v.p.russo@libero.it; 6Ufficio di Coordinamento Medico Legale, Istituto Nazionale Previdenza Sociale (INPS), 00144 Rome, Italy; valerio.sciannamea@inps.it (V.S.); raffaele.migliorini@inps.it (R.M.); 7Department of Occupational Medicine, University of Rome “Tor Vergata”, 00133 Rome, Italy; luca.coppeta@ptvonline.it (L.C.); andrea.magrini@uniroma2.it (A.M.)

**Keywords:** atrial fibrillation, healthcare costs, occupational health, medico-legal evaluation

## Abstract

Background: The aim of this research was to analyze trends in social security applications in Italy as a result of the onset of atrial fibrillation, analyzing data pertaining to the classification of professions and assessing the economic impact on the social security system. Methods: We analyzed all applications for invalidity allowances and invalidity pensions throughout Italy over a 10-year period from 01.01.2009 to 31.12.2019, giving specific attention to all reports indicating atrial fibrillation as the principal diagnosis (Cod. ICD-9-CM 427.31). We then extracted the relative expenditure data for said benefits. The results of all analyses have been collated in tables. Results: Over the period in question, a total of 3468 applications for assistance were filed throughout Italy indicating a diagnosis of atrial fibrillation, of which 58% were rejected, 41% qualified for an invalidity allowance, and only 1.1% qualified for a pension. On average, every year, 1100 workers received social security benefits as a result of a diagnosis of atrial fibrillation, which equates to an average annual expenditure of EUR 10 million. A comparison of the data from the first observation year (2009) with data from the last (2019) shows a rising trend in the number of beneficiaries and consequently in expenses. Conclusions: The social security assistance provided by the Italian government by means of the National Institute of Social Security is fundamental to social cohesion and to those who are either permanently disabled from working or those with a significantly diminished earning capacity. This assistance is associated with a significant financial cost, which requires careful monitoring.

## 1. Introduction

Cardiovascular diseases represent one of the most widespread categories of disease in the Western world, diseases that can compromise the health and physical strength, and consequently the working capacity, of those affected [1].

Atrial fibrillation (AF) is the most common sustained cardiac arrhythmia in adults, with an estimated prevalence of between 2% to 4%. A 2.3-fold rise is expected in the coming decades, largely owing to the extended longevity of the general population and intensifying search for undiagnosed AF [2]. AF patients have a five-fold increased risk of ischemic stroke or systemic embolism, and oral anticoagulation therapy should be considered when the CHA2DS2Vasc Score is ≥1 in men and ≥2 in women [3,4,5,6]. Moreover, AF is associated with substantial morbidity and mortality, thus portending a significant burden on patients, societal health, and health economy [7].

Atrial fibrillation can cause significant and permanent damage to health, resulting in the loss of working capacity, thus preventing those affected from continuing to be productive members of society [8].

Economic aid is established by law in order to provide financial support to those with a reduced or permanent loss of working capacity. Article 38 of the Constitution of the Republic of Italy states: “Every citizen unable to work and without the necessary means of subsistence is entitled to welfare support. Workers have the right to be assured adequate means for their needs and necessities in the case of accidents, illness, disability, old age and involuntary unemployment.

Disabled and handicapped persons are entitled to receive education and vocational training. Responsibilities under this article are entrusted to entities and institutions established by or supported by the State.

Private-sector assistance may be freely provided”.

To achieve said objectives, the National Institute for Social Security (INPS) was set up as the main social security agency of the Italian public pension plan. All public and private employees, along with the majority of self-employed workers who do not have their own social security fund, are required to register with INPS. These institutions receive social security contributions from employees through the companies by whom they are employed or from self-employed workers directly, calculated on the basis of income. INPS is regulated by the Ministry of Labor and Social Policies of the Republic of Italy and is responsible for assessing the conditions of those affected by disease and providing economic aid where appropriate.

All workers registered with INPS, in the event of an accident or chronic disease, have the right to avail themselves of one of the social security benefits provided by the law: the Invalidity Allowance (IA) or Invalidity Pension (IP). The qualifications for receiving such benefits are laid down by law.

In order to qualify for an invalidity allowance, the person must prove his or her invalid or unfit for work status is due to one or more infirmities or physical/mental defects, resulting in a working capacity reduced to less than one-third. IA is granted for a period of three years and can be confirmed for periods of the same duration. After three consecutive grants, the allowance is confirmed automatically. Those who receive such benefits may continue working, if able, at a reduced capacity.

To qualify for an invalidity pension, however, absolute and permanent working incapacity, that is, 100% invalidity, must be proved, caused by particularly severe conditions, such as advanced stage diseases, which result in serious and irreversible health conditions. In this case, the beneficiary ceases all working activity. Once granted, the invalidity pension is valid for life, subject to revision.

For both social security benefits, the applicant must satisfy certain administrative requirements: five years of INPS contributions, three of which must be within the five years preceding the date of application.

Clearly, a significant percentage of public finances at the Italian government’s disposal is used by INPS to provide such economic aid and must be subject to constant monitoring. If connected to diseases, the institute’s data may serve as an important source of information on public health and the nation’s future demographic, in addition to facilitating specific epidemiological, medico-legal, and occupational medicine assessments.

The objective of this study, which is a product of collaboration between teams of research, occupational health professionals, medico-legal experts, healthcare economists, and statisticians, is to analyze trends in applications and awards for the aforementioned benefits as a result of the onset of atrial fibrillation in accord with the various classifications of professions and to evaluate the economic impact of said economic aid on the social security system. This collection is completely unpublished on the international advertising panorama also due to the fact that the Italian legislation supports a particularly broad protection for workers. For this reason, more studies like this on different diseases are needed in order to understand their social, occupational, and economic impact on the system. These studies may be useful in order to better plan the use of economic and welfare resources, especially in the post-pandemic era.

## 2. Materials and Methods

In order to analyze the aforementioned social security benefits and subsequent estimates, we used the INPS national databases in cooperation with the INPS General Medico-legal Coordinator’s office. These databases contain benefit application reports made each year and the subsequent decisions made by the institute’s medico-legal commission, which include the principal diagnosis and a potential secondary diagnosis based on the International Classification of Diseases 9th Revision (ICD-9-CM). In fact, INPS medico-legal centers conduct an overall physical and mental health assessment of the applicant and then grant the request in the event of one or more invalidating diseases.

The study analyzes applications for Invalidity Allowance and Invalidity Pensions filed over a 10-year period from 01.01.2009 to 31.12.2019, focusing on all reports that indicate atrial fibrillation as the principal diagnosis (Cod. ICD-9-CM 427.31). We then extracted the relative expenditure data for said benefits. The results of all analyses have been collated in tables.

## 3. Results

Over the period in question, a total of 3468 applications for assistance were filed throughout Italy containing a diagnosis of atrial fibrillation (Table 1). A total of 58% of applications were rejected due to lack of the health requirements, 41% qualified for IA, and only 1.1% qualified for IP.

An average of 315 applications are filed every year, of which 184 are rejected, 127 qualify for IA, and four qualify for IP. Over the period in question, there has been an almost constant increase in the number of applications filed for this type of disease (Figure 1).

### 3.1. Analysis by Classification of Profession

As a result of the job descriptions contained in the reports, it was possible to conduct an analysis of distribution by classification of profession, selecting all applications granted during the period in question.

Table 2 shows the percentage distribution by classification of profession calculated on the basis of the number of applications for atrial fibrillation granted during the 10-year period. The classification of said work-related duties follows the Italian National Institute of Statistics (Istat) rules for the Classification of Professions, 2001 edition (Cp2001), which identifies eight macro groups of professionals:

I—Legislators, top management, and entrepreneurs. This group includes professions that require experience and specific decisional and organizational abilities. Duties comprise devising government policies, laws, and regulations on a local and national level; supervising implementation thereof; representing the State; directing, managing, and defining objectives; and providing direction for the activities performed by businesses and complex organizations and management structures.

II—Intellectual, scientific, and highly specialized professions. These include all professions that require a considerable level of knowledge—usually acquired by completing a graduate or post-graduate course of education—and experience in science, humanities, and art.

III—Technical professions. This group includes professions that require the operational expertise and necessary experience in order to perform support and technical–practical activities in the areas of science, humanities, social economics, sport, and art. The knowledge required for such professions can generally be acquired by completing a high school education or first level university course.

IV—Clerical support workers. This group includes non-management level office workers and non-manual executive work, which usually require the standard high school formal education.

V—Qualified professions in business activities and services. This group comprises professions that perform management/customer service roles in business activities, accommodation/catering services, recreational/family support/social services, and public security/personal and property protection services.

VI—Craftsmen, skilled workers, and farmers. This includes specialist manual labor professions in all areas of economic activity, requiring experience, knowledge of materials, tools, machines, production processes, and properties/potential uses of the final product.

VII—Plant operators, stationary and mobile machine operators, and vehicle drivers. This group includes the set-up, maintenance, and checking of machines, automated industrial plants, and automatic assembly lines of mass products, machines, or machine parts. It also includes vehicle drivers and mobile and hoisting machine drivers.

VIII—Unskilled professions. This group consists of professions that require a sufficient level of knowledge and degree of experience to perform very simple and repetitive activities which include using hand tools, often the use of physical force and limited autonomy in decisions and initiative. Professions that fall into this category perform unskilled manual labor in farming, industry, services, gardening, doorman, and cleaning. They also provide executive support in office work and small itinerant business-related tasks.

The results show that the most effected macro group is Group VI, craftsmen, skilled workers, and farmers (35%), followed by Group VIII, unskilled professions (27%). The least affected category is Group II, intellectual and scientific professions.

### 3.2. Territorial Analysis: Distribution by Region of Residence

Furthermore, it is possible to analyze the territorial distribution of applications filed for the diseases in question, since the applicant’s region of residence is also indicated. In order to exclude the effect of the various occupancy rates in the different regions, calculations were based on the ratio between the applications and the number of occupants in each region separately. The application distribution by region for 2009, 2019, and the average over the period in question is shown in Table 3.

In terms of the average number of applications filed over the 10-year period in relation to the average occupancy, Valle d’Aosta has a decidedly higher number of granted invalidity allowance and invalidity pension applications than the others. It thus qualifies for all intents and purposes as an outlier. High numbers are also reported in Molise, Basilicata, and Calabria, whereas the regions that report comparatively low numbers of accepted applications are Lombardy, Venice, and Piedmont.

### 3.3. Estimated Number of Beneficiaries and Social Security Costs

The total number of workers who receive social security benefits is published by the INPS Statistical Watchdog and displayed below in Table 4. Using the total number of beneficiaries per year, based on the same distribution percentage by disease of the total number of new applications, we calculated the number of beneficiaries affected by the disease in question, atrial fibrillation.

The amount awarded is dependent on the type of benefit (lesser for IAs and greater for IPs), the type of pension fund to which the worker belongs, and the sum of said worker’s contributions. Since information pertaining to individual workers is unavailable, the average monthly amounts for both social security benefits, shown in Table 4, has been used as a reference. The total social security burden for the disease in question has been estimated using the Monte Carlo simulation method [9].

The estimates show that each year, on average, approximately 1100 workers receive social security benefits due to a diagnosis of atrial fibrillation. The majority receive an invalidity allowance and as such can continue to work. These benefits cost approximately €10 million per year. A comparison of the data from the first observation year (2009) with the last (2019) shows a rising trend in the number of beneficiaries and consequently in expenses (Table 5).

## 4. Discussion

The results of the study show that cardiac diseases such as atrial fibrillation can lead to the partial or total loss of earning capacity for a significant number of people of working age. The fact that the Italian social security system is able to provide financial support for these types of workers is a considerable social victory that benefits the population, the value of which should neither be dismissed nor underestimated. For this reason, constant monitoring of social security payment data is essential, particularly for epidemiological and preventative purposes [10,11].

The Italian social security system spends in the region of EUR 11.5 million per year to support these workers. Over the 10 year period analyzed, this expenditure has increased by 50%. The fact that the age of the population of Italy has and continues [12] to increase significantly may be a contributory factor to said upsurge. This includes the ever-increasing percentage of the working age population that fall into the age bracket most associated with a higher incidence of cardiac disease.

An atrial fibrillation diagnosis can have significant repercussions on the fitness for work of those whose employment involves a high level of physical strength or creates specific risks (for example driving automobiles). In respect of driving, particularly when such involves the transport of third parties, there is concern over the risk of road traffic accidents caused by the onset of an arrhythmia. The statistics in the literature show, however, that such an event is infrequent: one recent study [13] showed that only 1.4/1000 non-fatal accidents and 4/1000 fatal accidents were ascribable to the driver’s health condition. Moreover, the most frequently identified clinical cause of such accidents was epilepsy, with all cardiac diseases accounting for just 8%.

Moderate physical activity does not seem to affect the onset of atrial fibrillation, whereas, according to a recent study, the risk of atrial fibrillation is significantly higher in those who work more than 55 h per week (OR: 1,44 1.42, 95% CI = 1.13–1.80, *p* = 0.003) [14,15,16].

It may also be possible that certain sectors are more affected by occupational risks such as work-related stress or instability. Further studies and analysis will need to be conducted in that regard.

The study also highlighted irregularities in the territorial distribution of benefits, taking into account the various occupancy rates in each region, and irregularities in the classifications of professions most affected by these diseases.

Despite such discrepancies, we believe that the social security system currently in place is functional and satisfies national needs. There is an evident need for constant monitoring for financial and healthcare purposes in addition to facilitating increasingly more uniform medico-legal assessments [17]. In our opinion, it is fundamental to provide ongoing training for and knowledge sharing between medical assessors and specialists in the various disciplines such as cardiology, occupational medicine, and legal medicine in order to identify emerging worker protection needs appropriately and accurately for the sustainability of the social security system and for the benefit of the nation.

Further studies should focus on other common diseases such as diabetes, asthma, pulmonary diseases, cardiovascular accidents, and strokes, etc., from the analysis of these data, useful elements could emerge in order to orient health policy in the field of prevention, budget disbursements due to new impacts of diseases, and projections of the impacts of diseases in the coming decades could be constructed.

## 5. Conclusions

The assistance provided by the National Institute of Social Security on behalf of the Italian government is fundamental to social cohesion and supports those permanently disabled from working or those with a significantly reduced earning capacity. The financial cost of providing said aid is considerable and must be carefully monitored. Over the last decade, there has been a significant rise in both applications for and subsequent grants of benefits due to atrial fibrillation diagnoses as a result of an increased average age in Italy and other European countries. It is likely that this upward trend will continue over the coming years.

## Figures and Tables

**Figure 1 ijerph-19-01883-f001:**
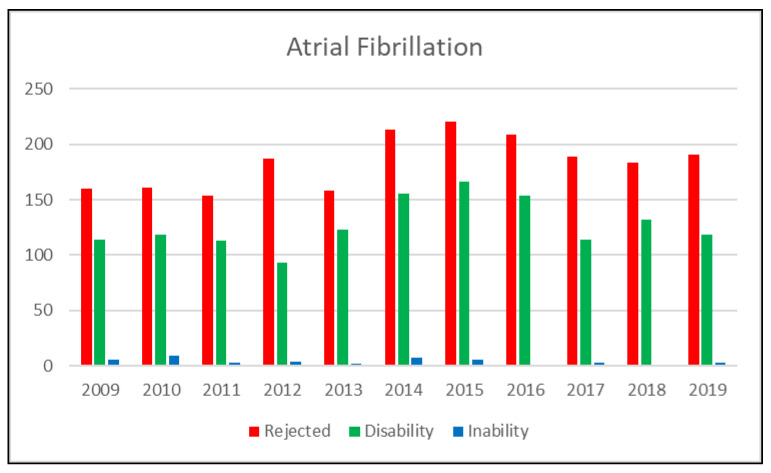
Total reports that indicate atrial fibrillation as the primary diagnosis, divided by decision and year. Absolute values. 2009–2019.

**Table 1 ijerph-19-01883-t001:** Total applications for each disease in absolute values.

		2009	2010	2011	2012	2013	2014	2015	2016	2017	2018	2019	Total
Atrial Fibrillation	Rejected	160	161	154	187	158	213	220	209	189	183	191	
	Disability	114	118	113	93	123	155	166	154	114	132	118	
	Inability	5	9	3	4	2	7	5	1	3	1	3	
	Total	279	288	270	284	283	375	391	364	306	316	312	3468

**Table 2 ijerph-19-01883-t002:** Distribution percentage by professional category of the applications granted for atrial fibrillation between 2009–2019.

Classification of Profession	Values %
I—Legislators, top management, and entrepreneurs	3.0%
II—Intellectual, scientific, and highly specialized professions	0.9%
III—Technical professions	3.2%
IV—Clerical support workers	4.6%
V—Skilled professions in business activities and services	11.3%
VI—Craftsmen, skilled workers, and farmers	35.1%
VII—Plant Operators, stationary and mobile machine operators, and vehicle drivers	14.5%
VIII—Unskilled professions	27.4%

**Table 3 ijerph-19-01883-t003:** Application distribution by region per 1000 occupants in 2009, 2019, and the average for the 10-year period. Values in percentages.

Region	2009	2019	Average 2009–2019
Piedmont	4.5%	0.8%	2.4%
Valle d’Aosta	0.0%	40.8%	27.2%
Liguria	4.8%	1.2%	3.2%
Lombardy	1.1%	0.5%	1.0%
Trentino-Alto Adige	0.0%	1.5%	3.0%
Venice	1.4%	0.0%	1.2%
Friuli-Venezia Giulia	3.0%	5.9%	4.0%
Emilia-Romagna	1.2%	3.3%	2.6%
Tuscany	3.4%	1.9%	2.6%
Umbria	6.3%	14.5%	12.3%
Marche	1.2%	2.4%	4.7%
Lazio	3.1%	2.8%	3.5%
Abruzzo	6.2%	7.6%	6.3%
Molise	0.0%	20.7%	18.8%
Campania	9.8%	9.1%	9.1%
Puglia	8.5%	10.4%	11.1%
Basilicata	20.0%	11.9%	12.9%
Calabria	13.1%	12.3%	12.7%
Sicily	4.6%	5.0%	7.2%
Sardinia	7.7%	6.4%	9.9%

**Table 4 ijerph-19-01883-t004:** Total social security benefits paid by year and average annual amounts.

Year	Total Beneficiaries		Average Annual Amounts	
	IA	IP	IA	IP
2009	366,728	81,138	€ 7835	€ 12,174
2019	446,326	84,183	€ 9593	€ 13,717
Media 2009–2019	392,913	84,106	€ 8765	€ 13,063

**Table 5 ijerph-19-01883-t005:** Estimated number of social security beneficiaries due to atrial fibrillation and related expenditure for 2009–2019. Absolute values.

	IA		IP		Total	
	Beneficiaries	Costs	Beneficiaries	Costs	Beneficiaries	Costs
2009	951	EUR 7,454,031	46	EUR 562,071	998	EUR 8,016,102
2019	1028	EUR 9,864,371	38	EUR 527,431	1067	EUR 10,391,802
AVERAGE	1060	EUR 9,331,781	44	EUR 10,391,802	1103	EUR 9,900,870

## Data Availability

Data are available on request from the corresponding author.

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
