# Peer review of "Health and Economic Impact of Atrial Fibrillation of Workers in Italy: Social Security Benefits"

_ijerph, 2022, doi:10.3390/ijerph19031883_

Round 1

Reviewer 1 Report

I am happy to review this interesting manuscript. Although this manuscript can stand the test of time, there are several weaknesses that should be addressed before it can be considered for possible publication.

Introduction: There are several short passages that should be combined to give a coherent presentation. Throughout the introduction and manuscript as well, there is hardly a single citation although the author(s) have included a reference list. The introduction failed to bring out the significance of the study and the existing gaps in terms of literature review and inconsistencies in the empirical findings.

Materials and methods: This section could be more detailed.

Results and discussion: Adequately discussed. The conclusions support the findings.

Overall contribution: Very limited in terms of theoretical contribution and practical implications.

Author Response

We want to thank the reviewer for his work and for helping us improve the manuscript. We have checked and modified the references and correctly indicated in the text. We have included other reasons why we believe the study we have done is important. We have also tried to make the introduction more understandable. We hope it can be adequate. Best regards

Reviewer 2 Report

The paper is well structured.

The author correctly used the existing literature and demonstrated the ability to use the methods.

Results are clearly presented.

I strongly suggest that the author should extend the conclusion, especially including clearly the policy implications and suggestions for further research.

Additionally, the introduction is brief and the author should be clear with their choices and implications of their research ( e.g. why the topic is important, why did they choose this country , why did they choose this time span).

I would also suggest to the author to consider comparing their results with previous studies.

Author Response

We want to thank the reviewer for his work and for helping us improve the manuscript. We have followed the indications and indications and added in the introduction the reasons on the basis of which work is important and useful in many fields. We then concluded in the final part of the considerations indicating the need for new studies and their reasons.

It was not possible to detect the literature of previous studies that evaluated the same data. thanks, we hope it can be adequate. Best regards

Reviewer 3 Report

I read closely the article. The article is well written, informative and interesting.   I'm persuaded that the onset of atrial fibrillation of workers in Italy constitutes an ongoing public health crisis with substantial government social security costs and other economic costs. The authors' conclusions make good common sense to me:

Author Response

We want to thank the reviewer for his work and for helping us improve the manuscript. We inserted the new title as recommended and made the required changes. Thanks again

Round 2

Reviewer 1 Report

I appreciate the authors for incorporating the suggestions and comments. The revised manuscript is improved in terms of quality and rigor. Good luck!